## THE NATURAL HISTORY OF MODEL ORGANISMS

# The unlimited potential of the great pond snail, *Lymnaea stagnalis*

**Abstract** Only a limited number of animal species lend themselves to becoming model organisms in multiple biological disciplines: one of these is the great pond snail, *Lymnaea stagnalis*. Extensively used since the 1970s to study fundamental mechanisms in neurobiology, the value of this freshwater snail has been also recognised in fields as diverse as host–parasite interactions, ecotoxicology, evolution, genome editing and 'omics', and human disease modelling. While there is knowledge about the natural history of this species, what is currently lacking is an integration of findings from the laboratory and the field. With this in mind, this article aims to summarise the applicability of *L. stagnalis* and points out that this multipurpose model organism is an excellent, contemporary choice for addressing a large range of different biological questions, problems and phenomena.

**ISTVÁN FODOR, AHMED AA HUSSEIN, PAUL R BENJAMIN, JORIS M KOENE[†] AND ZSOLT PIRGER[†]\***

## Introduction

In ancient Greece, over 2,400 years ago, it was already recognised that by studying animals we could learn much about ourselves. Over the centuries since then, it has become clearer that some species are highly suitable in the fields of medical, basic and applied biological research (*Ericsson et al., 2013*). However, when considered carefully, there is perhaps only a limited set of animal species that are versatile enough to lend themselves to become model organisms in multiple biological disciplines (*Frézal and Félix, 2015*; *Hilgers and Schwarzer, 2019*; *Markow, 2015*; *Phifer-Rixey and Nachman, 2015*).

In the second half of the 20th century, one booming line of research has focused on molluscs. Neuroscientists such as the Nobel Prize winners Alan Hodgkin, Andrew Huxley and Eric Kandel recognised these animals' potential as models for understanding basic neurobiological processes (*Hodgkin and Huxley, 1952*; *Kupfermann and Kandel, 1969*; *Wachtel and Kandel, 1967*). One particularly well-suited mollusc for this type of research is the freshwater pond snail, *Lymnaea stagnalis*, which has been used extensively since the 1970s to study the functioning of the nervous system from molecular signalling to behaviour.

The value of *L. stagnalis* also has been recognised in a wide range of applied biological fields. These include the study of host–parasite interactions, ecotoxicology, evolution, developmental biology, genome editing, 'omics' and human disease modelling. This extensive suitability stems from the most obvious advantages of *L. stagnalis*: its well-known anatomy, development (both of the embryonic and post-embryonic processes), and reproduction biology; its well-characterised central and peripheral nervous and neuroendocrine systems from key molecules to behavioural processes; and its readily accessible and mostly large neurons. There is also a growing body of available sequence data with an impending annotated genome and the option to use new technical approaches such as genome editing. Taking all of the above into consideration, these advantages simplify the study of different scientific topics integrated from the molecular to the population level.

**\*For correspondence:** pirger.zsolt@okologia.mta.hu

[†]These authors contributed equally to this work

**Competing interests:** The authors declare that no competing interests exist.

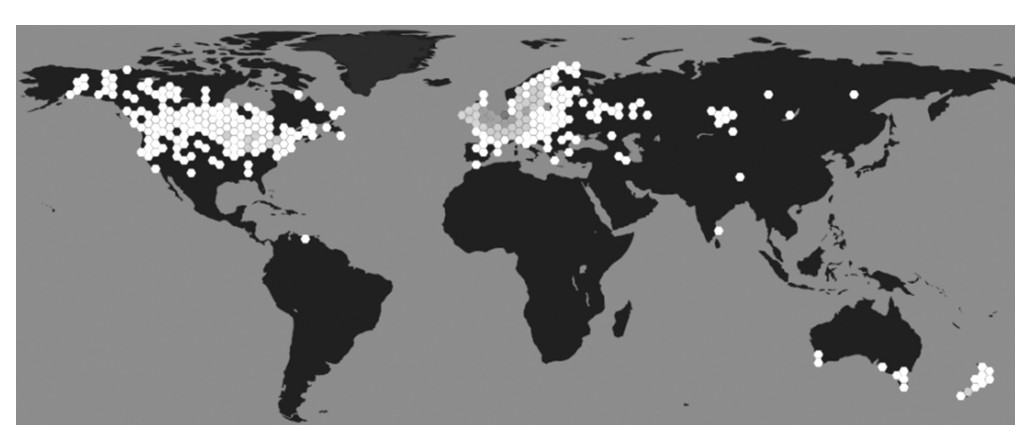

**Figure 1.** Geographical distribution of *L. stagnalis*. Places where this species of snail has been reported to occur (hexagons), shaded based on population density (white indicates low density and dark grey indicates high density; source data from *GBIF Secretariat, 2019*).

This article is a tribute to over 50 years of research with *L. stagnalis* that has resulted in a considerable contribution to the understanding of general biological processes. Here, we present the essential background information on the natural history of this freshwater snail. We also provide an overview of the ground-breaking and recent information on different research fields using *L. stagnalis* (and snails in general). Our aim is to showcase *L. stagnalis* as a contemporary choice for addressing a wide range of biological questions, problems and phenomena, to inspire more researchers to use this invertebrate as a model organism, and to highlight how findings from the laboratory and the field could be better integrated.

## Natural history of *L. stagnalis*

Initially described by Linnaeus in 1758 as *Helix stagnalis*, the species now known as *L. stagnalis* is generally referred to as the great pond snail (Panpulmonata; Hygrophila; Lymnaeidae). It is found throughout Northern America, Europe, and parts of Asia and Australia (*Atli and Grosell, 2016*; *Zhang et al., 2018a*; *Figure 1*). The snails inhabit stagnant and slowly running shallow waters rich in vegetation and are mainly herbivores, preferring algae, water plants and detritus (*Lance et al., 2006*). They are active all year round (even when there is a layer of ice on the water) but typically reproduce from spring to late autumn (*Nakadera et al., 2015*). They do not have a clear day-night rhythm, but display sleep-like behaviour (*Stephenson and Lewis, 2011*) and are more likely to lay eggs during daytime (*Ter Maat et al., 2012*). They are light to dark brown in colour and relatively large for pond snail species, with their spiral shells reaching lengths of up to 55 mm (*Benjamin, 2008*). In highly oxygenated water, they absorb oxygen directly across their body wall; but when dissolved oxygen levels drop, they switch to breathing via a lung accessed by a respiratory orifice called the pneumostome (*Lukowiak et al., 1996*).

*L. stagnalis* serves as the intermediate host for parasites including flatworms responsible for diseases such as fascioliasis (liver fluke and river rot) and cercarial dermatitis (swimmer's itch) in humans (*Adema et al., 1994*; *Davison and Blaxter, 2005*; *Ferté et al., 2005*; *Núñez et al., 1994*; *Skála et al., 2020*). Laboratory and field studies showed that penetration of a parasite into a snail will initiate a chronic infection in which the parasite alters snail neurophysiology, metabolism, immunity, growth and reproduction (*Kryukova et al., 2014*; *Langeloh and Seppälä, 2018*; *Vorontsova et al., 2019*). These studies have also investigated how selection acts on immune defence traits (*Langeloh et al., 2017*). Investigations of the natural history of *L. stagnalis*, which focus on host-parasite associations, aid the development of novel control measures that reduce snail-mediated parasitic transmissions. Primary predators of juveniles and adults include leeches, crayfish and fish, some of which snails can detect via chemicals that the predators emit (*Dalesman and Lukowiak, 2012*).

The life cycle and reproductive biology of the species are well-characterised (*Ivashkin et al., 2015*; *Koene, 2010*; *Mescheryakov, 1990*;

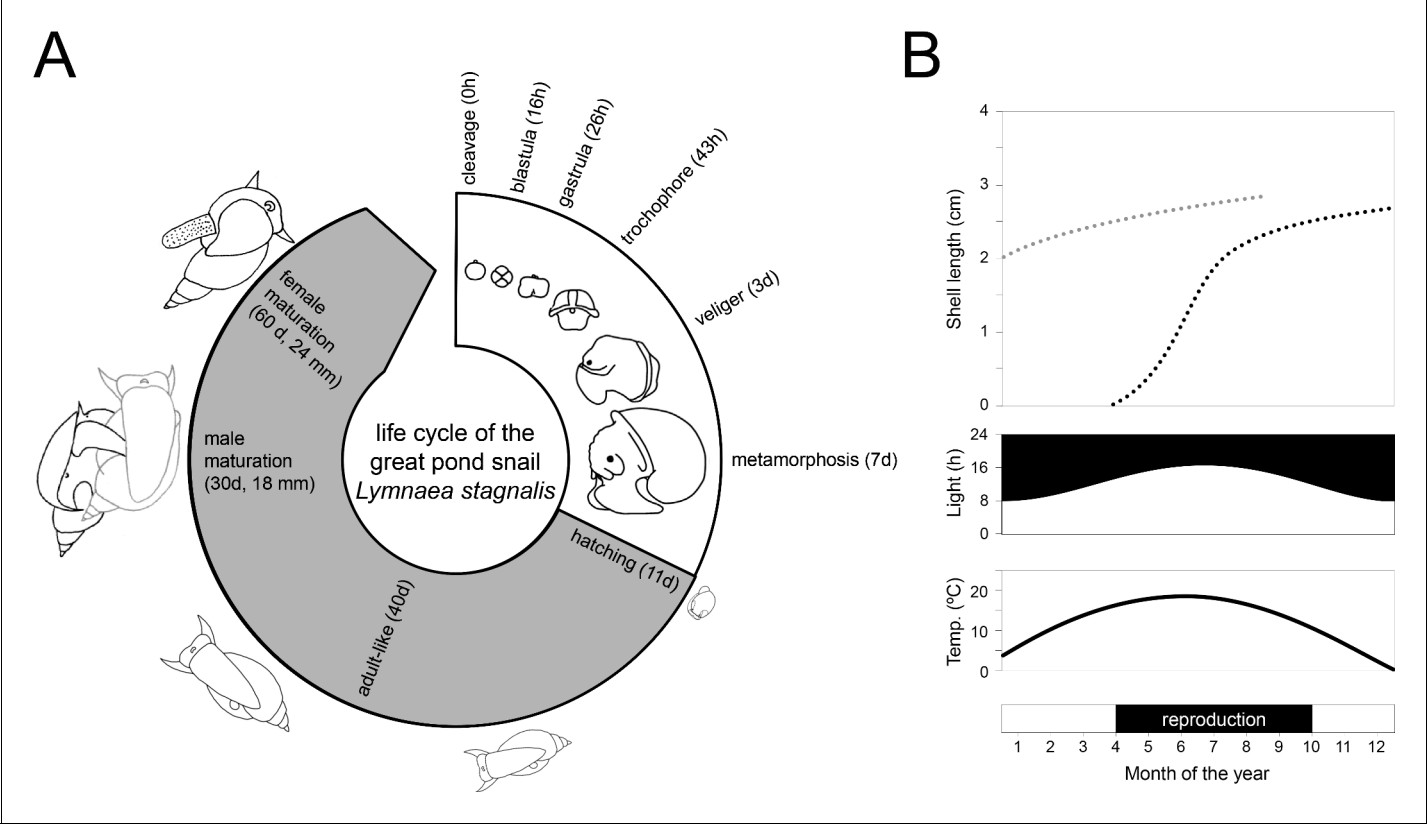

**Figure 2.** Life cycle and wild reproductive habit of *L. stagnalis*. (A) The embryonic development in the egg from zygote to hatching (over 11–12 days) is depicted in the white area of the life cycle and consists of six main stages: cleavage, blastula, gastrula, trochophore, veliger and metamorphosis (Source data from *Ivashkin et al., 2015*). The grey area of the life cycle depicts growth and development after hatching. Although *L. stagnalis* is a simultaneous hermaphrodite, the male reproductive organs are functional before the female ones (*Koene and Ter Maat, 2004*): specimens reach male and female maturation on average at an age of 30 and 60 days, respectively (based on *Koene, 2010*). (B) In the wild, generations only partly overlap, as depicted by the two dotted growth curves (top; based on *Nakadera et al., 2015*). Individuals that are born during spring and summer, overwinter as adults (light grey dotted line) after which they overlap with the adult generation of the next year (black dotted line). The external conditions such as light and temperature (middle), which strongly influence when egg laying occurs (bottom), are depicted for the situation in a typical temperate zone.

*Morrill, 1982*; *Figure 2A*). Embryos develop inside transparent eggs packaged in a translucent gelatinous mass allowing observers to follow their developmental stages in detail over the 11–12 days to hatching. The time from laid eggs to mature reproductive adults can be as short as two months depending on the temperature, photoperiod, feeding regime and mate availability at the location where they are being raised. In their natural habitat, they have been found to reach an age in excess of one year, but in the laboratory they live longer, up to two years (*Janse et al., 1988*; *Nakadera et al., 2015*). For laboratory breeding, a large and genetically diverse breeding stock is recommended as this will facilitate a well-standardised stock population without too much inbreeding. The largest and longest-maintained breeding facility is found at the Vrije Universiteit in Amsterdam, where *L. stagnalis* has been bred

continuously for over 50 years (*Nakadera et al., 2014*).

Its well-characterised embryonic and post-embryonic processes have promoted extensive use of *L. stagnalis* in the field of developmental biology. This snail has helped us to understand the mechanisms underlying shell formation (*Hohagen and Jackson, 2013*), the transfer of non-genetic information to the developing embryos (*Ivashkin et al., 2015*), and resource allocation during development (*Koene and Ter Maat, 2004*). Moreover, studies with *L. stagnalis* has also helped develop and evaluate models in physiology, such as the "dynamic energy budget" model (*Zonneveld and Kooijman, 1989*; *Zimmer et al., 2014*).

This snail is a simultaneous hermaphrodite, meaning that mature individuals express a functional male and female reproductive system at the same time within one body. Despite having

## Box 1. Outstanding questions about the natural history of the great pond snail.

- Why is inbreeding depression less strong in *L. stagnalis* than in related freshwater snail species?
- How different are long-term laboratory-bred strains from natural populations as a result of different selection pressures influencing development, mating propensity, self-fertilisation, learning and/or changes in sensitivity due to changing biotic and abiotic factors?
- How can the knowledge about host-parasite interaction be applied to control the spread of parasites in the natural habitat?
- How phenotypically plastic or evolutionarily adaptable is this species to changes in biotic and abiotic conditions in its habitat (e.g., temperature, light and/or chemical pollution, and resulting changes in ecosystem composition)?
- Why are sinistral individuals not found more often in natural populations and what does that mean for the natural selection pressures on this chiral morph?
- Are the detection and avoidance of positive and negative stimuli only present in the laboratory or is this learned behaviour also exhibited under field conditions (e.g., predicting presence of food, mating partners and/or predators)?
- How can the knowledge about the regulatory mechanisms underlying reproduction be better used to understand the evolution and flexibility of the hermaphroditic lifestyle?

two functional sexes, specimens copulate unilaterally; one individual plays the male role and the other the female role within one mating interaction (*Hoffer et al., 2017*). There is no obligate alternation of sexual roles, but when both individuals of a mating pair are motivated to mate in the male role they can perform a second copulation with the same partner in the opposite sexual role (*Koene, 2010*; *Ter Maat et al., 2012*; *Van Duivenboden and Maat, 1985*). An integrated laboratory and field study showed that, in the wild, most individuals are born during the spring and summer seasons and generations partly overlap because the latter cohorts overwinter and overlap with mature individuals of the new spring cohort (*Figure 2B*; *Nakadera et al., 2015*). The same study showed that both age and size significantly affected the sex role decision under laboratory conditions. This species is quite fecund in the laboratory: snails from the mass culture in Amsterdam produce a large number of offspring all year round (*Nakadera et al., 2014*); however, an initial field study found a more moderate fecundity rate in natural populations (*Nakadera et al., 2017*). In the laboratory, specimens produce on average 2–3 egg masses per week each containing 100–150 eggs, depending on the body size of the individual (*Nakadera et al., 2014*). The hatching

rate under laboratory conditions is generally above 90% (*Hoffer et al., 2017*). Based on laboratory, semi-field and field studies, explicit inbreeding or self-fertilisation depression for this species have been found to be absent (*Coutellec and Lagadic, 2006*; *Escobar et al., 2011*; *Koene et al., 2008*; *Puurtinen et al., 2007*) or very unlikely (*Coutellec and Caquet, 2011*), however the reasons for this remain unclear (*Box 1*). Nevertheless, eggs are preferentially outcrossed with sperm from mating partners, which can be stored for two months, and individuals only use their own 'autosperm' when this 'allosperm' is not available (*Nakadera et al., 2014*).

## A gold standard model organism for neuroscience

The squid *Loligo forbesii* and sea hare *Aplysia californica* were the first molluscan models for examining neuronal processes. *L. stagnalis* emerged shortly afterwards, and was described as "a reductionistic, yet sophisticated model to address fundamental questions in learning and memory" (*Rivi et al., 2020*). Molluscs were used extensively in the field of neurobiology in the 20th century, typically because their central nervous systems were, in most cases, more

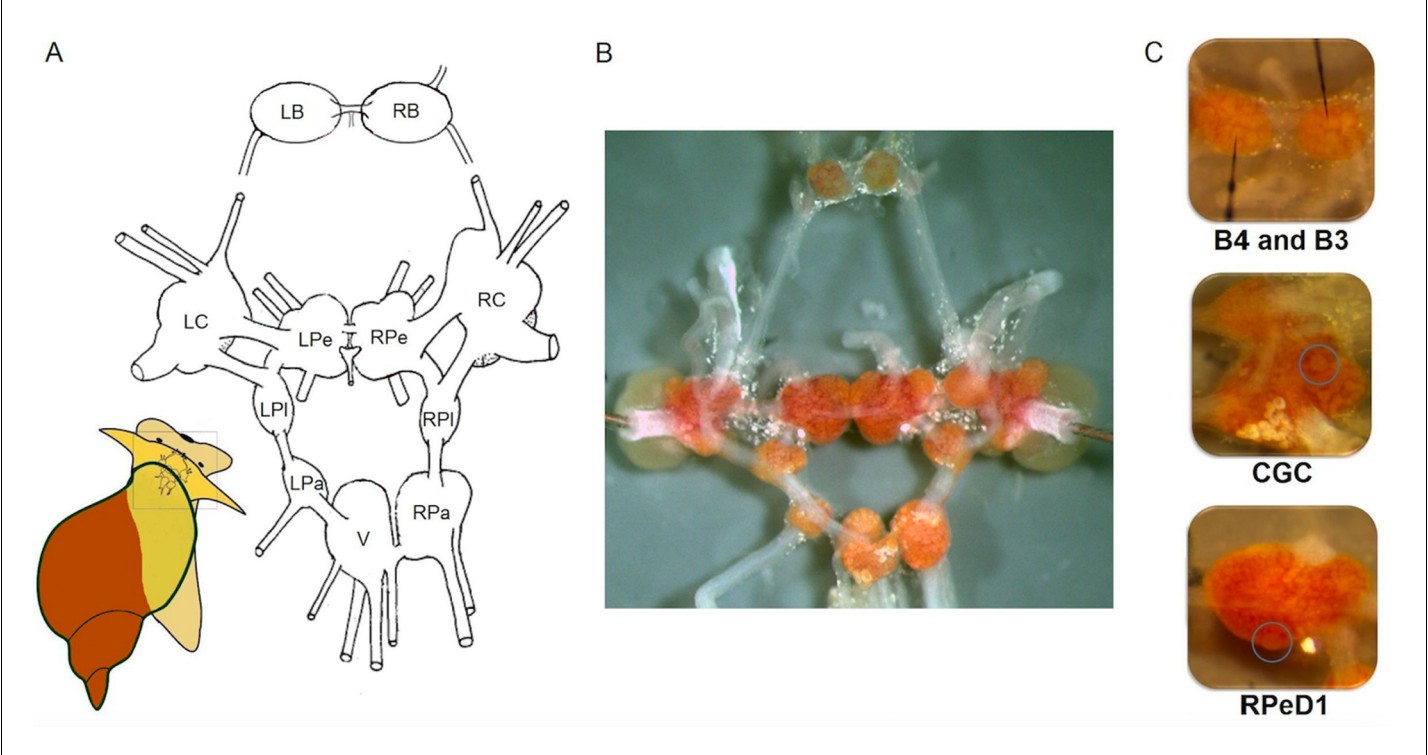

**Figure 3.** The central nervous system and identified single neurons of *L. stagnalis*. (A) Schematic map (dorsal view) of the isolated whole central nervous system that is formed of the paired (left and right) buccal (LB, RB), cerebral (LC, RC), pedal (LPe, RPe), pleural (LPl, RPl), parietal (LPa, RPa) and unpaired visceral (V) ganglia. (B) Isolated central nervous system showing the arrangement of the 11 interconnected ganglia. Brightly pigmented orange-coloured neurons are localised on the surfaces of the ganglia. (C) Identified single neurons: B4 (left), B3 (right; motor neurons responsible for the implementation of feeding), CGC (interneuron in cerebral ganglia modulating the feeding and learning) and RPeD1 (interneuron in pedal ganglia regulating the respiration and heartbeat).

accessible than those of vertebrate animals. Technical developments since then mean many such experiments can now be performed on vertebrates as well, yet we would argue that invertebrates still have substantial advantages for our understanding of the central nervous system.

The relatively simple central nervous system of *L. stagnalis* is organised in a ring of 11 interconnected ganglia (*Figure 3A,B*) with ~25,000 neurons. The neurons are mostly large (30–150 µm in diameter) and their bright, orange-coloured cell bodies are located on the surface of the ganglia (*Figure 3B*; *Kemenes and Benjamin, 2009*). Thus they are readily accessible for experimental purposes, simplifying investigations of neural clusters, circuits and even single neurons, which can be reliably identified for functional examination using a variety of approaches such as electrophysiological, molecular and analytical techniques (*Crossley et al., 2018*; *de Hoog et al., 2019*; *El Filali et al., 2015*; *Harris et al., 2012*; *Kemenes et al.,*

*2011*; *Lu et al., 2016*; *Samu et al., 2012*; *Wagatsuma et al., 2005*; *Zhang et al., 2018b*).

Individual neurons (*Figure 3C*; *Benjamin and Crossley, 2020*) and their synaptic connectivity were identified as parts of circuits controlling behaviours (*Audesirk et al., 1985*; *Benjamin, 2012*; *McCrohan and Benjamin, 1980a*; *McCrohan and Benjamin, 1980b*; *Syed and Winlow, 1991*; *Syed et al., 1990*). Combining this knowledge with an understanding of the molecular mechanisms, often from laboratory studies, has helped produce an integrated picture of the processes underlying learning and memory, such as consolidation, reconsolidation, extinction and forgetting. The molecular pathways involved in memory formation in *L. stagnalis* were recently identified, providing further evidence the mechanisms of learning and memory consolidation are conserved across phylogenetic groups in a variety of learning paradigms, including non-associative or associative learning, and operant or classical conditioning (*Benjamin and Kemenes, 2013*; *Fulton et al.,*

## Box 2. How can findings at different biological levels be integrated to better understand this species' natural history?

- It needs to be established at what level *L. stagnalis* can function as a model for medical research such as neurodegenerative disease and be a substitute for standard vertebrate models. This requires a better understanding of how such functions affect this species in its natural habitat.
- The new molecular techniques and available 'omics' data provide an incentive for research that aims to understand the mechanisms underlying natural history processes such as sex allocation, simultaneous hermaphroditism, reproductive success, chirality and learning.
- The knowledge about learning and decision-making in the laboratory needs to be extended to field populations to promote future developments in, for example, neural network-inspired robotics.

*2005*; *Josselyn and Nguyen, 2005*; *Kemenes and Benjamin, 2009*; *Kemenes et al., 2002*; *Marra et al., 2013*; *Michel et al., 2008*; *Nikitin et al., 2008*; *Park et al., 1998*; *Pirger et al., 2010*; *Pirger et al., 2014a*; *Pirger et al., 2014b*; *Ribeiro et al., 2003*; *Rivi et al., 2020*; *Sadamoto et al., 1998*; *Sadamoto et al., 2010*; *Schacher et al., 1988*; *Vigil and Giese, 2018*; *Wan et al., 2010*). Recently studies have also revealed differences in learning ability at the behavioural level between situations in the laboratory and the field (e.g., *Dalesman and Lukowiak, 2012*; *Dalesman et al., 2015*; *Dalesman, 2018*).

The well-characterised proximate processes at the molecular, cellular and circuit levels mean studying this simple nervous system has the potential to provide insights into how snails can respond appropriately to environmental challenges (e.g., climatic change or pharmacologically active compounds). Also, since their behaviours are generated by reflexive and central pattern generator networks similar to those of vertebrates (*Katz and Hooper, 2007*), results from snails offer insights into the fundamental processes important for these animals too.

Finally, recent developments have enabled this species to be used as a model for understanding the basis of neurodegenerative diseases. Comparative analyses have yielded several homologs to human genes linked to ageing and neurodegenerative diseases in *A. californica* and this species has proved well suited for studying these processes (*Moroz et al.,*

*2006*; *Moroz and Kohn, 2010*). Similar molecular sequences have been identified in *L. stagnalis* (*Fodor et al., 2020b*). With the appropriate genetic background, its accessible central nervous system and relatively long and well-characterised life span mean *L. stagnalis* is highly suitable for studying the biological mechanisms of ageing, age-related memory loss and neurodegenerative diseases, such as Parkinson's and Alzheimer's diseases (*Arundell et al., 2006*; *de Weerd et al., 2017*; *Ford et al., 2017*; *Hermann et al., 2007*; *Hermann et al., 2020*; *Maasz et al., 2017*; *Patel et al., 2006*; *Pirger et al., 2014b*; *Scutt et al., 2015*; *Vehovszky et al., 2007*; *Yeoman and Faragher, 2001*; *Yeoman et al., 2008*).

### Ecotoxicology and risk assessment in a changing global environment

It has become clear that in the globalised world, climate change, light pollution, micro- and nanoplastics, and pharmacologically active compounds all pose a challenge to animal life. These challenges affect the availability of suitable habitats and reduce the quality of the land, lakes and rivers. They also change the environmental composition of pathogens, parasites, competitors and invaders. Understanding how global ecosystems are adapting to pollution is a complex problem; it requires researchers to monitor natural populations and conduct laboratory studies to discover the bases of adaptations or the lack thereof (*Markow, 2015*).

Molluscs the second most diverse animal group and considered to be excellent indicators of ecosystem health. For example, *L. stagnalis* is a sensitive and reliable species for such studies (*Amorim et al., 2019*), in a large part because of its well-characterised developmental and reproductive biology as described above. The major targets in this field of study have been metal-risk assessment (*Crémazy et al., 2018*; *Pyatt et al., 1997*; *Vlaeminck et al., 2019*), the effects of pesticides (*Coutellec et al., 2008*; *Lance et al., 2016*; *Tufi et al., 2015*; *Vehovszky et al., 2015*), nanotoxicology (*Hudson et al., 2019*; *Stoiber et al., 2015*), the development of toxicokinetic models (*Baudrot et al., 2018*), immunocompetence analyses (*Boisseaux et al., 2018*; *Gust et al., 2013b*), and global warming risk assessment (*Leicht et al., 2017*; *Leicht and Seppälä, 2019*; *Teskey et al., 2012*). Studies on *L. stagnalis* have measured toxicological values such as mortality concentrations (e.g., LC50) and impairment of reproduction (e.g., EC50) but also sub-lethal and more sensitive endpoints such as reproductive success, growth, cellular and molecular biomarkers that may be coupled with behavioural responses (*Amorim et al., 2019*).

*L. stagnalis* has also been recognised as a useful organism to examine the effects of pharmacologically active compounds and micro- and nanoplastics on aquatic organisms (*Amorim et al., 2019*; *Charles et al., 2016*; *Ducrot et al., 2014*; *Gust et al., 2013a*; *Horton et al., 2020*; *Pirger et al., 2018*; *Zrinyi et al., 2017*). However, it must be pointed out that researchers need to be critical of the generalisability of results while performing such experiments since there are differences between the endocrine system of molluscs and vertebrates; molluscs, for example, do not have functional oestrogen receptors (*Eick and Thornton, 2011*; *Lagadic et al., 2007*; *Scott, 2012*). It is also important to recognise that molluscs are not suitable for some types of ecotoxicological studies and they cannot always substitute for fish.

Notably, *L. stagnalis* is the first aquatic non-arthropod invertebrate model organism to be recognised in environmental risk assessments. The developed standard reproduction test was officially approved by the national coordinators of the Organisation for Economic Cooperation and Development (*OECD, 2016*) thus paving the way for investigating ecotoxicological effects in more detail. Such information will contribute to a more complete picture of the mode of action of potentially toxic substances and other environmental factors and provide assessments of risk for individual species of different types and wider ecosystems.

## Combining evolution and natural history

Within the field of evolutionary biology, *L. stagnalis* has helped us to understand the evolution of several phenomena. Left-right asymmetry is a general evolutionary phenomenon seen across a variety of species, including humans where the congenital condition situs inversus results in the mirrored position and shape of the heart and liver (*Blum et al., 2014*; *Oliverio et al., 2010*; *Palmer, 2009*; *Palmer, 2016*). The coil or chirality of snail shells is one of the more spectacular outward manifestations of this asymmetry. Snails found in nature can have shells that coil either to the right or left, with most species being right-coiling. Specimens of *L. stagnalis* that coil in the opposite left-winding, or sinistral, direction are rare and often categorised as 'unlucky' because their different chirality makes it difficult for them to mate the more usual, right-winding individuals (*Davison et al., 2009*). Left-winding snails are also less able to learn in a mate-choice context (*Koene and Cosijn, 2012*). The existence of the two different morphs has made this species ideal for studying chiromorphogenesis, i.e. the first step of left-right symmetry breaking. Genes and signalling pathways that are responsible for snail coiling have been identified (*Abe et al., 2014*; *Davison et al., 2016*; *Kuroda, 2014*; *Kuroda, 2015*; *Kuroda et al., 2016*), and similar signalling pathways are required for vertebrate chiromorphogenesis as well (*Kuroda, 2015*). Studies on *L. stagnalis* can give important insights into the evolution of body plans in other phyla, and may have wider medical implications, including an understanding of situs inversus.

*L. stagnalis* has also played a crucial role in studies into the evolution of hermaphroditism and its consequences for sexual selection. This area of research relies heavily on a solid understanding of the natural history of this species. Selection of sexual traits that affect mating success was previously considered not to act in simultaneous hermaphrodites (*Charnov, 1979*; *Darwin, 1871*; *Greeff and Michiels, 1999*; *Morgan, 1994*). However, recent research, including work with *L. stagnalis*, has contradicted this earlier conclusion (*Anthes et al., 2010*; *Baur, 1998*; *Chase, 2007*; *Janicke et al., 2016*; *Michiels, 1998*; *Nakadera and Koene, 2013*).

Unilateral mating of *L. stagnalis* offered a unique opportunity to test whether sexual selection acts independently on the two sexual roles of a simultaneous hermaphrodite (*Anthes et al., 2010*; *Arnold, 1994*; *Hoffer et al., 2017*). Recent experiments have also revealed that male and female reproductive strategies can be optimised independently in this species. This was done by measuring sexual selection gradients (also called Bateman gradients), which reveal the relationship between the number of matings and the reproductive success of the sexual functions (*Anthes et al., 2010*). Experiments with *L. stagnalis* showed that this mating system seems largely male-driven, and that the sexual selection gradients are consistently positive for the male function but change over time to benefit the opposite sex (*Anthes et al., 2010*; *Hoffer et al., 2017*; *Janicke et al., 2016*; *Pélissié et al., 2012*). These pioneering works, which measured and quantified the processes of sexual selection and their underlying mechanisms, thus incorporated this hermaphrodite into the general Darwin-Bateman paradigm that had so far mainly been tested on separate-sexed species. They also described both the evolutionary potential and limitations of hermaphrodite animals and revealed important practical applications for the conservation of wildlife.

## New opportunities from a growing multi-omics coverage

From about 1980, continued attention was given to the physiological characterisation of *L. stagnalis*, but more recent research has focussed on an 'omics' approach to better understand the underlying molecular processes (*Santama et al., 1993*; *Santama et al., 1995a*; *Santama et al., 1995b*; *Santama and Benjamin, 2000*). Due to its pre-eminence as a model system in

**Table 1.** List of some of the most important (neuro)peptides identified in *L. stagnalis*.

| Molecule | Abbreviation | Function | Accession number | Reference |
|---|---|---|---|---|
| caudodorsal cell hormones | CDCH | reproduction | P06308 | *Vreugdenhil et al., 1988* |
| FMRFamides | FMRF | reproduction, cardiac control | P19802 | *Linacre et al., 1990* |
| conopressin | - | reproduction | AAB35220 | *Van Kesteren et al., 1995* |
| neuropeptide Y | NPY | reproduction, development | CAB63265 | *De Jong-Brink et al., 1999* |
| actin-related diaphanous genes (1, 2) | dia 1, dia 2 | development, chirality | KX387869, KX387870 KX387871, KX387872 | *Kuroda et al., 2016* |
| insulin-related peptides (I, II, III, V, VII) | MIPs | development | CAA41989; P25289; AAB28954; AAA09966; AAB46831 | *Smit et al., 1991*; *Smit et al., 1992*; *Smit et al., 1993b*; *Smit et al., 1996*; *Smit et al., 1998* |
| sodium stimulating hormone | SIS | ion and water control | P42579 | *Smit et al., 1993a* |
| small cardioactive peptide | SCP | feeding, cardiac control | AAC99318 | *Perry et al., 1999* |
| myomodulin | MIP | feeding, cardiac control | CAA65635 | *Kellett et al., 1996* |
| pituitary adenylate cyclase-activating polypeptide-like molecule | PACAP-like | learning and memory | - | *Pirger et al., 2010* |
| cAMP response element-binding proteins (1, 2) | CREB 1 CREB 2 | learning and memory | AB041522; AB083656 | *Sadamoto et al., 2004* |
| glutathione reductase and peroxidase | Gred Gpx | metabolic detoxification | FJ418794, FJ418796 | *Bouétard et al., 2014* |
| catalase | CAT | metabolic detoxification | FJ418795 | *Bouétard et al., 2014* |
| superoxide dismutase | SOD | metabolic detoxification | AY332385 | *Zelck et al., 2005* |
| heat-shock protein | HSP70 | stress response | DQ206432 | *Fei et al., 2007* |
| molluscan defence molecule | MDM | immune system | AAC47132 | *Hoek et al., 1996* |
| allograft inflammatory factor-1 | AIF-1 | immune system | DQ278446 | *van Kesteren et al., 2006* |

neuroscience, early molecular studies tended to focus on the central nervous system (*Feng et al., 2009*; *Johnson and Davison, 2019*). The favourable anatomical features enabled the accumulation of peptidomic data from the mass spectrometry of single neurons (*Perry et al., 1999*; *Worster et al., 1998*), making the neuropeptidergic system the most intensely studied part of the central nervous system (*Buckett et al., 1990*; *Perry et al., 1998*). Taking advantage of a variety of platforms available for nucleotide sequencing: Sanger (*Davison and Blaxter, 2005*; *Sadamoto et al., 2004*; *Swart et al., 2019*), Illumina (*Korneev et al., 2018*; *Sadamoto et al., 2012*; *Stewart et al., 2016*), BGISEQ (*Jehn et al., 2018*) and Oxford Nanopore (*Fodor et al., 2020a*), many sequencing methodologies have been successfully applied to this species.

Extensive genomic, transcriptomic and peptidomic data for *L. stagnalis* are available in the NCBI database. Four major transcriptome datasets were established by sequencing mRNA from the central nervous system (*Bouétard et al., 2012*; *Davison and Blaxter, 2005*; *Feng et al., 2009*; *Sadamoto et al., 2012*), and then used to identify genes and proteins, thus providing a solid genetic background for *L. stagnalis*. Furthermore, an unannotated draft genome is already available and a collaborative effort is underway to produce an annotated genome (*Johnson and Davison, 2019*) which would largely solve the problem of the lack of molecular information that has so far inhibited research in the *L. stagnalis* model system (*Rivi et al., 2020*). Approximately 100 (neuro)peptides have been identified so far (*Benjamin and Kemenes, 2020*), encoded by genes involved in various regulatory processes (*Table 1*). These findings contributed to a global understanding of the natural history of *L. stagnalis* by characterising the molecular and cellular processes underlying chirality, reproduction, immune processes, host-parasite interaction, and acute and chronic adaptive responses to toxic substances in the environment.

Furthermore, the CRISPR/Cas9 genome editing method has recently been applied to molluscs (*Henry and Lyons, 2016*; *Perry and Henry, 2015*). In *L. stagnalis,* it was used to knock out the gene responsible for coiling direction during development, leading to a better understanding of chirality in the life of the two morphs (*Abe and Kuroda, 2019*). The establishment of genome editing in *L. stagnalis* opens up significant opportunities for functional genomics

to investigate the role of specific genes, for example, in snail developmental, toxicology and immunobiological studies.

## Conclusion

Research on model organisms has been essential to developing the current understanding of how life works. The unique features of *L. stagnalis* make it an excellent experimental system to complement the classic invertebrate (*C. elegans, D. melanogaster*) and vertebrate (*D. rerio, M. musculus*) models. Research utilising this species is expected to lead to future breakthroughs in a number of scientific fields, especially in neuroscience and evolutionary biology. For example, as a simultaneously hermaphroditic outcrossing species, it presents the opportunity to test the generality of hypotheses that are mainly based on non-hermaphroditic or self-fertilising models. There is considerable information about the natural history of *L. stagnalis* compared to some other model species, but we feel some areas of research using *L. stagnalis* – in particular neurobiology and ecotoxicology – would benefit by extending more of their studies out of the laboratory and into the field. We believe that a deeper integration of information from field studies with input from laboratory findings – such as applying experimental designs and approaches developed in the laboratory to populations in the wild – will provide future opportunities for further innovation (*Box 2*). Such efforts could address the unanswered questions regarding this model organism (see *Box 1*). Significantly, emerging recent technical approaches such as pocket-sized sequencing devices, especially with their impending breakthrough also in protein sequencing, start allowing researchers to perform more experiments in the field such as following molecular mechanisms of learning.

**István Fodor** is in the NAP Adaptive Neuroethology, Department of Experimental Zoology at the Balaton Limnological Institute, Centre for Ecological Research, Tihany, Hungary

**Ahmed AA Hussein** is in the Department of Ecological Sciences, Faculty of Sciences at Vrije Universiteit, Amsterdam, the Netherlands

**Paul R Benjamin** is at Sussex Neuroscience, School of Life Sciences, University of Sussex, Brighton, United Kingdom

**Joris M Koene** is in the Department of Ecological Sciences, Faculty of Sciences at Vrije Universiteit, Amsterdam, the Netherlands

https://orcid.org/0000-0001-8188-3439

Zsolt Pirger is in the NAP Adaptive Neuroethology, Department of Experimental Zoology at the Balaton Limnological Institute, Centre for Ecological Research, Tihany, Hungary

pirger.zsolt@okologia.mta.hu

https://orcid.org/0000-0001-9039-6966

*Author contributions:* István Fodor, Conceptualization, Writing - original draft, Writing - review and editing; Ahmed AA Hussein, Paul R Benjamin, Conceptualization, Writing - original draft; Joris M Koene, Conceptualization, Visualization, Writing - original draft; Zsolt Pirger, Conceptualization, Supervision, Funding acquisition, Visualization, Writing - original draft, Writing - review and editing

*Competing interests:* The authors declare that no competing interests exist.

### Funding

| Funder | Grant reference number | Author |
|---|---|---|
| National Brain Research Project | 2017-1.2.1-NKP-2017-00002 | Zsolt Pirger |

The funders had no role in study design, data collection and interpretation, or the decision to submit the work for publication.

### Decision letter and Author response

Decision letter https://doi.org/10.7554/eLife.56962.sa1
Author response https://doi.org/10.7554/eLife.56962.sa2

## Additional files

### Data availability

All data generated during the preparation of this review are included in the manuscript.

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
