## [Decision Letter]

Thank you for submitting your article "The Natural History of Model Organisms: The unlimited potential of the great pond snail, *Lymnaea stagnalis*" for consideration by *eLife*. Your Feature Article has been reviewed by three peer reviewers, and the evaluation has been overseen by an Associate Features Editor (Stuart King). All individuals involved in the review of your submission have agreed to reveal their identity: Iker Irisarri (Reviewer #1); Elena Voronezhskaya (Reviewer #2); Marie-Agnes Coutellec (Reviewer #3).

After a consultation with the reviewers, the editor has drafted this decision to help you prepare a revised submission. We appreciate that the COVID-19 pandemic is disrupting many aspects of everyday life, so please take as long as you need to work on these revisions.

The editor may also contact you separately about some editorial issues that you will need to address.

Summary:

This essay is being considered as part of a series of articles on "The Natural History of Model Organisms": https://elifesciences.org/collections/8de90445/the-natural-history-of-model-organisms. Each article should explain how our knowledge of the natural history of a model organism has informed recent advances in biology, and how understanding its natural history can influence/advance future studies.

This article highlights the main advantages of the freshwater snail (*Lymnaea stagnalis*) as a model organism for research in various scientific domains. The authors recall the specific features that have led this species to be used as an invertebrate model for the past 50 years and identify pending issues in these domains. Broadening the scope of previous reviews on this organism, details of *L. stagnalis's* natural habitat and life history are also presented in a clear and compact format.

Overall, the reviewers felt the authors had done a good job of summarizing a broad body literature on the topic and had judiciously selected information to provide an engaging account of this model mollusc. This article may interest a wide scientific community across various fields, and particularly any researcher wishing to start using *L. stagnalis* as a de novo model.

Below are a number of details that should be attended to prior to publication.

Essential revisions:

1) Strengthen connection to primary literature:

There is a tendency throughout the article to cite previous reviews. It would be more useful for readers – and for those researchers actively working with this model organism – if the citations were instead to the original research as much as possible.

In particular, the reviewers suggest citing the relevant primary literature for some of the basic information: e.g., the species distribution (currently Amorim et al., 2019), breathing physiology (currently Benjamin, 2008), diet (currently Lance et al., 2008), or reproductive period (currently Nakadera et al., 2015).

2) Benefits as a model organism:

The Introduction notes that: "The value of *L. stagnalis* also has been recognized in a surprisingly-wide range of applied biological fields". The Introduction would also benefit if the main features that make L. staganalis a valuable model could be quickly summarised in a single sentence. In general, giving the evidence to support the assertions as to the value of this model helps to avoid parts of the text being perceived as dogmatism. Related to this, and while there was no doubt that the species is suited to test many hypotheses, there was a feeling among the reviewers that softening some claims and doing more to acknowledge that other models may be more relevant or have other specific advantages (depending on the issue or the state of knowledge at the start of the project) would be beneficial.

3) Highlight connections between field and laboratory studies:

The reviewers also noted that most of the article reports on laboratory experiments, which likely reflects an existing bias in the literature. Yet given that the aim of this article (and the wider collection) is to provide a context of natural history, more reporting on studies in natural settings would be preferable. In the conclusions, the authors noted: "[A deeper integration of 'field' information with input from laboratory] will allow for novel experimental designs and approaches developed in the laboratory to be applied to field populations and/or in the field". Other explicit examples of how the integration between laboratory and field experiments would be the key to success would strengthen the article. This would also be one way that the authors could help to address the "lack of integration between laboratory and field experiments" that they recognize as a major missing element in *L. stagnalis* research.

4) Highlight contributions to diverse fields:*Lymnaea* has had a long history in the field of developmental biology, and, in the field of physiology, where it was also one of the first model species to which the dynamic energy budget (DEB) theory was applied (see Zonneveld, C., & Kooijman, S. A. L. M. (1989). Application of a dynamic energy budget model to *Lymnaea stagnalis* (L.). Functional Ecology, 269-278). It would be good if these details could be briefly mentioned at the relevant points in the text.

---

## [Author Response]

Essential revisions:1) Strengthen connection to primary literature:There is a tendency throughout the article to cite previous reviews. It would be more useful for readers – and for those researchers actively working with this model organism – if the citations were instead to the original research as much as possible.In particular, the reviewers suggest citing the relevant primary literature for some of the basic information: e.g., the species distribution (currently Amorim et al., 2019), breathing physiology (currently Benjamin, 2008), diet (currently Lance et al., 2008), or reproductive period (currently Nakadera et al., 2015).

Unfortunately, in some places, the relevant primary literatures had to be removed from the text and had to be substituted with a review paper to comply with the word count. However, we agree with the reviewers and made some changes in the relevant parts. To note, Nakadera et al., 2015 is not a review, but a research study having integrated the situation of the lab and field, and so we kept this paper.

2) Benefits as a model organism:The Introduction notes that: "The value of L. stagnalis also has been recognized in a surprisingly-wide range of applied biological fields". The Introduction would also benefit if the main features that make *L. stagnalis* a valuable model could be quickly summarised in a single sentence. In general, giving the evidence to support the assertions as to the value of this model helps to avoid parts of the text being perceived as dogmatism. Related to this, and while there was no doubt that the species is suited to test many hypotheses, there was a feeling among the reviewers that softening some claims and doing more to acknowledge that other models may be more relevant or have other specific advantages (depending on the issue or the state of knowledge at the start of the project) would be beneficial.

We agree with the reviewers and added the asked summary to the Introduction about the main features that make *L. stagnalis* a valuable model species in a wide range of biological fields. Moreover, we added some parts to the main text (Neuroscience and Ecotoxicology sections) that compare the relevance and usefulness of *L. stagnalis* to other models (mice, fish) for some types of studies.

3) Highlight connections between field and laboratory studies:The reviewers also noted that most of the article reports on laboratory experiments, which likely reflects an existing bias in the literature. Yet given that the aim of this article (and the wider collection) is to provide a context of natural history, more reporting on studies in natural settings would be preferable. In the conclusions, the authors noted: "[ A deeper integration of 'field' information with input from laboratory] will allow for novel experimental designs and approaches developed in the laboratory to be applied to field populations and/or in the field". Other explicit examples of how the integration between laboratory and field experiments would be the key to success would strengthen the article. This would also be one way that the authors could help to address the "lack of integration between laboratory and field experiments" that they recognize as a major missing element in *L. stagnalis* research.

We are grateful to the reviewers for pointing this out. We made an effort to include and discuss additional field studies in the manuscript and in some places we made a comparison between the findings of the laboratory and field. Furthermore, we modified and supplemented the Conclusion.

4) Highlight contributions to diverse fields:Lymnaea has had a long history in the field of developmental biology, and, in the field of physiology, where it was also one of the first model species to which the dynamic energy budget (DEB) theory was applied (see Zonneveld, C., & Kooijman, S. A. L. M. (1989). Application of a dynamic energy budget model to Lymnaea stagnalis (L.). Functional Ecology, 269-278). It would be good if these details could be briefly mentioned at the relevant points in the text.

There was a short paragraph in the original text about the contribution of *L. stagnalis* to the field of developmental biology, however, it was removed to comply with the word count. At the same time, we agree with the reviewers and added the suggested paragraph to the main text (Introduction and Natural history sections).